# Multivalent binding kinetics resolved by fluorescence proximity sensing

Clemens Schulte[1], Alice Soldà[2], Sebastian Spänig[3], Nathan Adams[4], Ivana Bekić[4], Werner Streicher[4], Dominik Heider[3], Ralf Strasser[2] & Hans Michael Maric [1✉]

Multivalent protein interactors are an attractive modality for probing protein function and exploring novel pharmaceutical strategies. The throughput and precision of state-of-the-art methodologies and workflows for the effective development of multivalent binders is currently limited by surface immobilization, fluorescent labelling and sample consumption. Using the gephyrin protein, the master regulator of the inhibitory synapse, as benchmark, we exemplify the application of Fluorescence proximity sensing (FPS) for the systematic kinetic and thermodynamic optimization of multivalent peptide architectures. High throughput synthesis of +100 peptides with varying combinatorial dimeric, tetrameric, and octameric architectures combined with direct FPS measurements resolved on-rates, off-rates, and dissociation constants with high accuracy and low sample consumption compared to three complementary technologies. The dataset and its machine learning-based analysis deciphered the relationship of specific architectural features and binding kinetics and thereby identified binders with unprecedented protein inhibition capacity; thus, highlighting the value of FPS for the rational engineering of multivalent inhibitors.

[1] Rudolf Virchow Center, Center for Integrative and Translational Bioimaging, University of Wuerzburg, Josef-Schneider-Str. 2, Germany, 97080 Wuerzburg, Germany. [2] Dynamic Biosensors GmbH Germany, Lochhamer Strasse 15, 82152 Martinsried/Planegg, Germany. [3] Department of Bioinformatics, Faculty of Mathematics and Computer Science, Philipps-University of Marburg, Hans-Meerwein-Strasse 6, 35043 Marburg, Germany. [4] Nanotemper Technologies GmbH, Flößergasse 4, 81369 Munich, Germany. ✉email: Hans.Maric@uni-wuerzburg.de

Protein-protein interactions (PPIs) are of fundamental importance for cellular function and dysfunction[1] with up to 40% of all PPIs involving short, linear motifs located in intrinsically disordered protein regions[2]. Targeting and probing such PPIs contributes substantially to our understanding of physiology, pathology and ultimately the identification of novel pharmacological strategies[3]. The development of selective PPI modulators with high target protein binding affinities is facilitated by biophysical technologies that enable the determination of binding parameters of large binder libraries with minimal sample requirements. In particular, multimeric or branched peptides[4–6] provide superior binding specificities and affinities due to avidity[7, 8] by exploiting protein homo-oligomerization[9] observed for more than half of all proteins[10], offering enormous potential for the design of multivalent drugs, including novel drug modalities such as trivalent PROTACs[11], They commonly exhibit slower off-rates, and thus enhanced residence times, compared to their monovalent counterparts[12]. Despite the availability of robust theoretical mechanistic frameworks[8], the accurate prediction of multivalent binding dynamics based on biophysical properties of the interactors alone remains challenging. This is especially true for systems where higher valencies and complex, heterogeneous topologies occur or where structural information is incomplete. Vice versa, systematic experimental structure-activity relationship studies remain scarce, mainly due to laborious workflows, commonly relying on sequential synthesis, multimerization and labelling, necessitating multiple re-purification steps of the often comparably large compounds.

The engineering of multivalent architectures benefits from kinetic methodologies, such as surface plasmon resonance (SPR) or biolayer interferometry (BLI)[13–15]. However, such surface-based techniques are vulnerable to artefacts that result from the comparably high affinities and slow off-rates of multivalent binders, causing re-binding and, depending on immobilisation density, interference from neighbouring proteins through crosslinking[8].

Here we demonstrate the use of Fluorescence Proximity Sensing (FPS) as an alternative approach to study multivalent peptide-protein interactions in high-throughput and its value for effectively decoding higher order multivalent structure-activity relationships and thereby facilitating the guided engineering of such interactions.

## Results

FPS detects the binding of molecules in real-time through changes in the dye's local environment[16]. FPS, based on SwitchSENSE technology, relies on a biochip with covalently attached single stranded anchor DNA for target protein immobilization at a distance of approximately 30 nm[17], thereby potentially precluding re-binding and avidity effects. The peptide (analyte) binding is reported by a fluorescent reporter close to the immobilized protein of interest (ligand) (Fig. 1) and consequently independent of unspecific binding of the analyte to the chip surface. Importantly, FPS neither requires direct fluorescent labelling of the ligand nor the analytes, thereby avoiding dye-mediated artefacts. In contrast to other recently reported kinetic methods[18], FPS allows for the analysis of slow ($<10^{-4}$ s$^{-1}$) off-rates and fast on-rates ($>10^6$ M$^{-1}$s$^{-1}$).

While the workflow is designed to be applicable to any multivalent system where combinatorial display is feasible, we here use the neuronal scaffolding protein gephyrin[19] (geph) and its structurally resolved[20] interactor, the glycine receptor (GlyR) β subunit. PPIs within receptor protein complexes[21] and specifically scaffolds of the neuronal synapses are explored with multivalent chemical probes[22–26] and studied in pharmacological

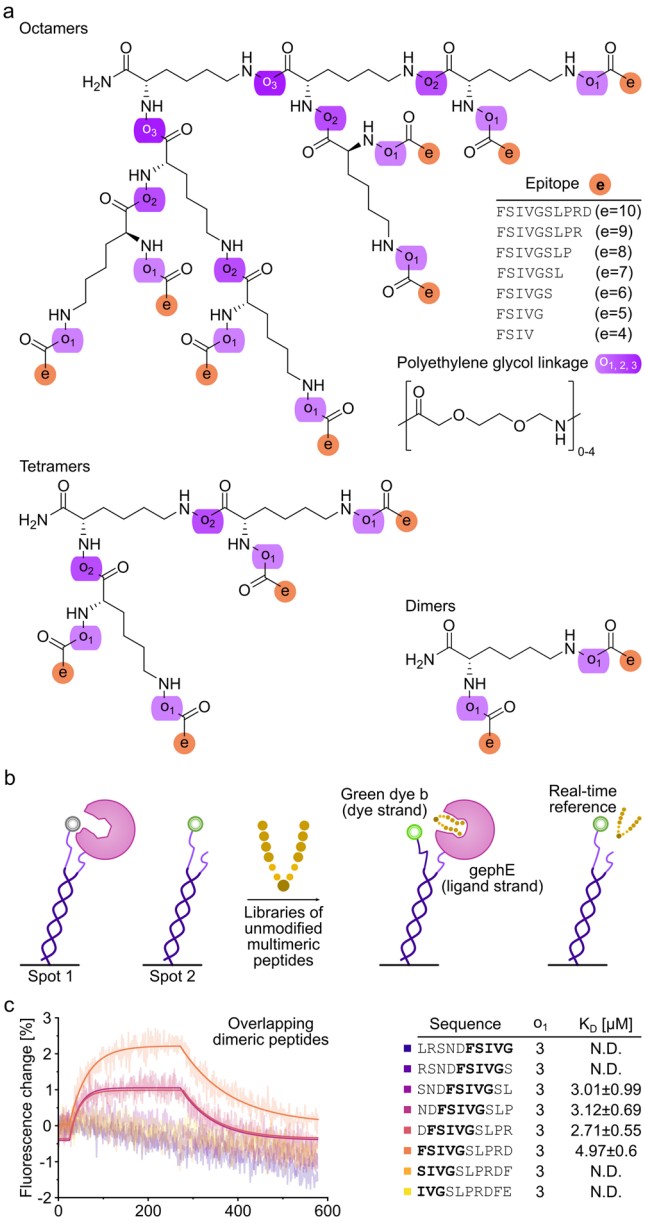

**Fig. 1 Multivalent peptide architectures, FPS setup and PPI mapping.** **a** Architecture of multimeric geph-binding peptides. An (Fmoc)-L-Lys(Fmoc) building block facilitated multimerization of geph-binding epitopes, linked together by PEG moieties ($o_{1-3}$), yielding dimeric, tetrameric and octameric peptides. **b** schematic representation of FPS measurements. The receptor-binding domain of the neuronal scaffolding protein gephyrin (gephE) is immobilized on the ligand strand via an NHS coupling. The binding of unmodified, multimeric peptides during the association phase is detected by a change in fluorescence of green dye b. Note that a real-time reference on spot 2 is used to control for unspecific binding or influence on the fluorophore (**c**) Real-time affinity determination of overlapping, dimeric GlyR β derived peptides in FPS. Peptides were used at a concentration of 1 μM. Note that only peptides with a centred FSIVG core binding motif exhibited a measurable affinity.

Figure labels (panel a): Octamers; Epitope e; Epitope sequences: FSIVGSLPRD (e=10), FSIVGSLPR (e=9), FSIVGSLP (e=8), FSIVGSL (e=7), FSIVGS (e=6), FSIVG (e=5), FSIV (e=4); Polyethylene glycol linkage ($o_{1,2,3}$); Tetramers; Dimers.

Figure labels (panel b): Spot 1; Spot 2; Libraries of unmodified multimeric peptides; Green dye b (dye strand); gephE (ligand strand); Real-time reference.

Panel c table:

| Sequence | $o_1$ | $K_D$ [μM] |
|---|---|---|
| LRSND**FSIVG** | 3 | N.D. |
| RSND**FSIVGS** | 3 | N.D. |
| SND**FSIVGSL** | 3 | 3.01±0.99 |
| ND**FSIVGSLP** | 3 | 3.12±0.69 |
| D**FSIVGSLPR** | 3 | 2.71±0.55 |
| **FSIVGSLPRD** | 3 | 4.97±0.6 |
| **SIVG**SLPRDF | 3 | N.D. |
| **IVG**SLPRDFE | 3 | N.D. |

context (ClinicalTrials.gov, NCT04689035)[27]. Dimeric, tetrameric and octameric binders were synthesized using an accessible and broadly applicable strategy by combining binding sequences with Polyethylene glycol (PEG) linkers and L-Lysine cores[28] as branching points (Fig. 1a). Using varied geph binding sequences, PEG linkers of variable length and up to three branching points,

we synthesized a total of +100 unique multimeric compounds (Supplementary Table 1), differing over one magnitude in molecular weight.

For the FPS measurements, the otherwise unlabelled receptor binding geph E-domain (gephE) was coupled to the ligand strand while a fluorescent reporter was attached to the dye strand (Fig. 1b) To ensure that the structural integrity and dimeric composition of gephE is not compromised by the linkage to the ligand strand, the hydrodynamic status of the immobilized protein was measured in dynamic mode (Supplementary Fig. 1) and suggested that gephE is immobilized as a dimer. The functional integrity of immobilized gephE was further validated by comparing it to two non-binding gephE point variants (Supplementary Fig. 2)[29]. Among six tested dyes, the fluorescence change was highest for the green dye B (Dynamic Biosensors GmbH, DE) (Supplementary Fig. 3) which was therefore used in all subsequent FPS measurements. The functionality of this setup was demonstrated by recapitulating the structurally resolved geph-binding site of GlyRβ ($^{398}$FSIVG$^{402}$)[20] using a 1 μM library of unmodified, overlapping dimeric peptides with an offset of one amino acid (Fig. 1c). In all FPS measurements, the baseline fluorescence change in absence of gephE and a blank measurement without peptide was used as a control (Supplementary Fig. 4).

**Comparison of FPS with ITC, BLI and TRIC.** Next, we assessed the reliability of apparent $K_D$ values determined in FPS (Fig. 2a and Supplementary Fig. 5) by comparing this setup with commonly used immobilization- and in-solution-based PPI-quantification methods. Namely, real-time binding quantification using biolayer interferometry (BLI), high-throughput temperature related intensity change (TRIC) quantification as well as precise calorimetric measurements (ITC) (Fig. 2a). Compared to ITC measurements, which can be considered the gold standard as they quantify directly and label-free in solution (Fig. 2a and Supplementary Fig. 6), high-throughput quasi label-free TRIC measurements (Fig. 2a and Supplementary Fig. 7) recapitulate the same trend. The only exception being compound $e = 8$, $o_1 = 0$, $o_2 = 4$, which was outside of the dynamic range. The ITC measurements further confirm the expected binding mode of two gephE proteins per dimer and four gephE proteins per tetramer. The BLI measurements (Fig. 2a) necessitated loading densities and ligand concentrations that prevented effective dissociation of tetramers (Supplementary Fig. 8) and octamers (Supplementary Fig. 9). Thus, affinities could not be derived from single curves but were instead assessed through steady-state BLI measurements using multiple peptide concentrations (Supplementary Fig. 10). The determined $K_D$ values only partly recapitulated the affinities determined in ITC, possibly due to avidity effects such as re-binding.

Along the same line, BLI overestimated the affinity of the tetramers and further enabled the measurement of $e = 5$, $o = 2$, a lower affinity dimer. Conversely, the on- and off-rates of dimeric peptides were resolvable in BLI (Supplementary Fig. 11). However, a poor signal-to-noise ratio (SNR) was observed for small dimeric peptides (Supplementary Fig. 11c, d). In stark contrast, FPS enabled measurements of dimeric, and tetrameric compounds independent of compound size (Fig. 2a). The resolved binding hierarchy is in line with ITC and TRIC, similar to the apparent dynamic range.

Next, we compared the protein sample consumption of the four different biophysical PPI quantification methods (Fig. 2b). In terms of target protein consumption by weight, FPS performed second best among the methods employed, consuming 28.5-fold less protein than BLI measurements for sensor functionalization (0.64 μg for one FPS sensor chip versus 18.25 μg for 8 BLI

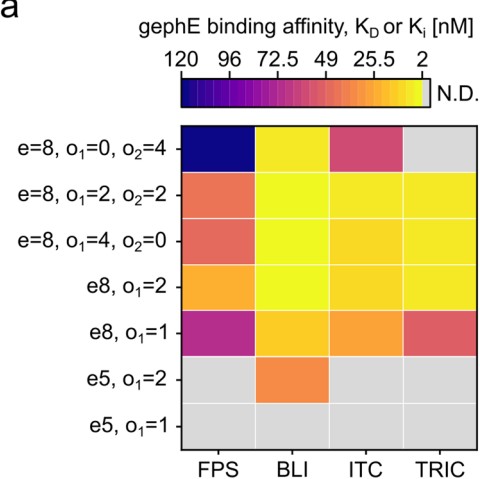

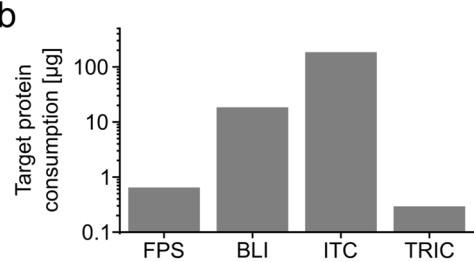

**Fig. 2 Comparison of apparent affinities of dimeric and tetrameric peptides in ITC, TRIC, BLI and FPS and target protein consumption.**
**a** Apparent binding affinities of seven benchmark peptides (four dimeric and three tetrameric) were measured using FPS (dynamic $K_D$ value from $n = 1$ measurements of 3 different concentrations), BLI (steady-state $K_D$ value from $n=1$ measurements at seven different concentrations), ITC ($K_D$ value from $n = 3$ and $n = 2$ measurements) and TRIC ($K_i$ values from $n = 2$ measurements). For complete measurements, see Supplementary Fig. 5, 6, 7 and 10 respectively. **b** Amount of target protein consumed for affinity determination of one peptide in the four methods tested (FPS: one sensor chip functionalization, BLI: functionalization of eight sensors, ITC: one run with 16 injections, TRIC: 16-point dose response in displacement assay setup). Source data are provided in Supplementary Data 1.

biosensors), 285-fold less than ITC (182.4 μg for one run) and 2.2-fold more than TRIC (0.29 μg for a 16-point dose response). Note that the amount of target protein consumed per FPS measurement may vary depending on e.g. the labelling efficiency.

To facilitate the determination of kinetic binding parameters of hundreds of peptides with a short turnover, we explored the possibility to directly couple FPS to low μM scaled solid-phase peptide synthesis. Consequently, we determined the intra-synthesis reproducibility of real-time affinity measurements of multimeric, unmodified peptides in FPS. $K_D$ values and kinetic parameters could be determined with low deviation using independently synthesized dimers and tetramers (Supplementary Fig. 12), indicating that the combined setup allows for reproducible and precise kinetic interactions studies.

**High-throughput determination of protein affinities and kinetics using FPS.** Next, we used the established FPS setup to resolve the relationship between multimeric peptide architecture and binding kinetics. Specifically, an array of dimeric, tetrameric, and octameric compounds was subjected to FPS measurements at a fixed concentration of 1 μM to achieve sufficient signal amplitude for weaker binders (Fig. 3). In addition to the on- and off-rates determined from functions fit to the obtained curves,

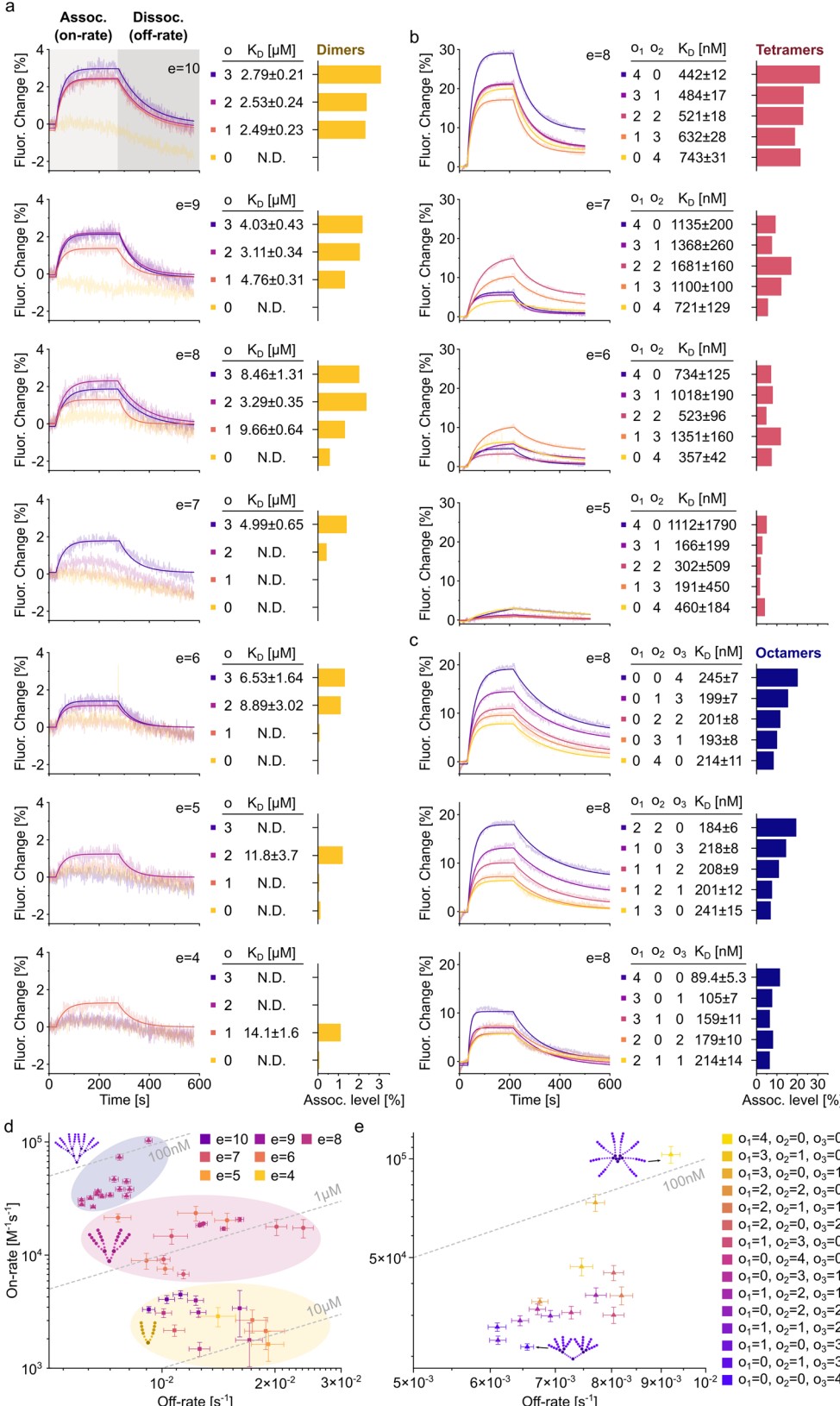

association levels, at which the measured curves during the association phase plateaued, were determined for each peptide. Overall, a prominent gain in affinity could be observed from dimers (Fig. 3a, low µM) to tetramers (Fig. 3b, high nM) and finally octamers (Fig. 3c, mid/low nM). Indeed, plotting of the obtained on-rates against the off-rates for each compound in a

rate-map (Fig. 3d) reveals that multimer affinity primarily depends on the valency. This is in line with previous studies[8] which demonstrated that increased valency also increases the ability to create additional binding conformations within the configurational network. The second most important factor is the length of the epitope. This trend recapitulates the changes in

**Fig. 3 FPS resolves binding kinetics of dimeric, tetrameric, and octameric peptide binders in high-throughput. a–c** FPS curves of all dimers (**a**, yellow), tetramers (**b**, red), and octamers (**c**, purple) tested are displayed next to the respective association levels. Peptide architecture is denoted as described in Fig. 1a). In brief, epitope length is denoted by "e" and the length of the respective PEG linker is represented by $o_1$, $o_2$ or $o_3$. For a complete list of the kinetic parameters of all compounds tested, refer to supplementary table 1. **d** Rate map of all dimers (yellow), tetramers (red), and octamers (blue) with a determinable on- and off-rate. Epitope lengths are color-coded. Note the the high dependence of dimer affinity on epitope length. 10 µM, 1 µM, and 100 nM affinities are indicated as dashed, grey lines. **e** Zoomed-in view of (**d**) with octameric binders in focus. Varying architectures are color-coded, and 100 nM affinity is indicated as dashed, grey line. Note that octamers with highest affinity contain ≥3 PEG building blocks in the outer $o_1$ position. Error bars represent fit uncertainty in $n = 1$ measurement. Source Data are provided as Supplementary Data 1.

binding strength that have been observed for the respective monovalent counterparts (Maric et al.[30]). In the here studied multivalent system, the observed affinity gain is primarily driven by on-rate effects which vary over two magnitudes, while the off-rates vary only 5-fold across all tested species. Together, these data confirm the importance of the binding affinity of the single binding epitopes for higher valency systems, demonstrating the importance of on-rate effects.

**FPS correlates multivalent topology and binding dynamics**. To resolve how topological multimeric features determine on- and off-rates, our measurements included a series of compounds identical in epitope length and number but systematically varied scaffold arrangement. Plotting the obtained on-rates against the off-rates for each compound as a rate-map, together with color-coding of the topological adjustments visualizes a clear trend (Fig. 3e). The octamer with the lowest affinity is characterized by a multivalent architecture that enables flexible movement of the two sides of the multimer but sterically restricted movement of the epitopes themselves within the two tetramers. Vice versa, the multimeric architecture that enabled the greatest flexibility close to the epitopes while at the same time enforcing pre-orientation of the epitopes through sterical constrains in the centre displayed the highest affinity. The difference in affinity between both compounds is primarily driven by on-rate (4.5-fold) but also off-rate effects (1.4-fold). This dataset resolves the structure-activity relationship of multivalent geph-binders and provides a framework for the development high-valency, ultra-high affinity interactors in general.

**Prediction of multivalent binding parameters**. The 40 successfully measured compounds constitute only a small fraction of the theoretical possible combinations. To discern whether the obtained dataset allows to predict multimer properties, we used machine learning. Specifically, we applied the Random Forest Regressor using the encoded amino acids and analogous building blocks as training input. Here, the peptide sequences are represented through the amino acid composition[31], which demonstrated overall good performance across multiple applications and provides easy interpretability[32]. First, we explored whether the observed on- and off-rates and the resulting $K_D$ values can be reliably predicted. To this end, we applied a leave-one-out cross-validation and found a high correlation between predicted and observed $K_D$ values (Fig. 4a and Supplementary Table 2), off-rates (Fig. 4b) and on-rates (Fig. 4c) in case of the tetrameric and octameric group. In case of the dimeric peptides, a positive correlation was only found for the $K_D$ values. We additionally examined the correlation between observed association level and $K_D$, on- and off-rate for each compound. Here, positive Pearson correlations were found in case of the dimeric group for $K_D$ (Fig. 4d) and especially off-rate (Fig. 4e) but not on-rate (Fig. 4f). In stark contrast, no or even negative correlations were found in the tetramer and octamer group when correlating the observed

association level to the $K_D$ values (Fig. 4d), off- (Fig. 4e) and on-rates (Fig. 4f).

Taken together, these results indicate that for both lower avidity dimers and higher avidity tetramers and octamers, $K_D$ values can be reliably predicted across multivalent species using the outlined algorithm. In stark contrast, the association level may only be a representative metric for $K_D$ and off-rate for distinct topology classes.

**Peptide binders with high avidity potently neutralize native gephyrin**. Our FPS studies suggest that higher-order geph-binding multimers possess enhanced potency as inhibitors compared to their dimeric counterparts. Using a complementary peptide microarray-based approach[33] with native geph from mouse brain lysates, we probed the geph neutralizing capacity of dimeric and tetrameric geph binders. Native geph was pre-incubated with dimeric, tetrameric and octameric binders (Fig. 4g) with varying architecture at increasing peptide competitor concentrations. Reduction in on-chip peptide binding by geph thus corresponds to neutralization of geph by competitor binding. Tetrameric and octameric binders exhibited up to two orders of magnitude more potent geph neutralization than the dimeric binders (Fig. 4h), thereby confirming the outcome of the FPS-based high-throughput screen and further highlighting the value of the outlined approach for avidity-based binding optimization.

**Discussion**

FPS is a versatile technique for measuring binding affinities of different binder–ligand systems with strongly varying size and composition of the binding partners, usefull to resolve complex and multiphasic binding events[34], commonly DNA/protein[16, 35], protein/protein[34] and protein/small molecules[36]. This study employs FPS in tandem with automated, low µM-scaled solid-phase peptide synthesis to establish a platform for high-throughput real-time binding affinity determination. This setup was used to systematically characterize +100 multimeric peptides with varying architecture, binding to the target protein geph. Contrary to other examples of kinetic studies of multimeric binders[37], we observed an increase in binding affinity of higher-order multimers mainly driven by an increase in on-rate. Our work confirmed valency and monovalent binding affinity as the primarily relevant design features that govern the magnitude of avidity enhanced binding. In the same line, we found that within the complex octameric linker architecture, a high degree of flexibility close to the geph-binding epitope enhances affinity as opposed to high flexibility within the core of the octamer. This observation could be explained by an improved preorientation and/or access to additional binding conformations. Further, we demonstrate the successful data-based prediction of affinities, currently hard to achieve using biophysical and structural data alone.

Major limitations of contemporary kinetic methods such as BLI are irresolvable off-rates in case of high avidity compounds (Supplementary Figs. 5 and 6). Gratifyingly, the here presented

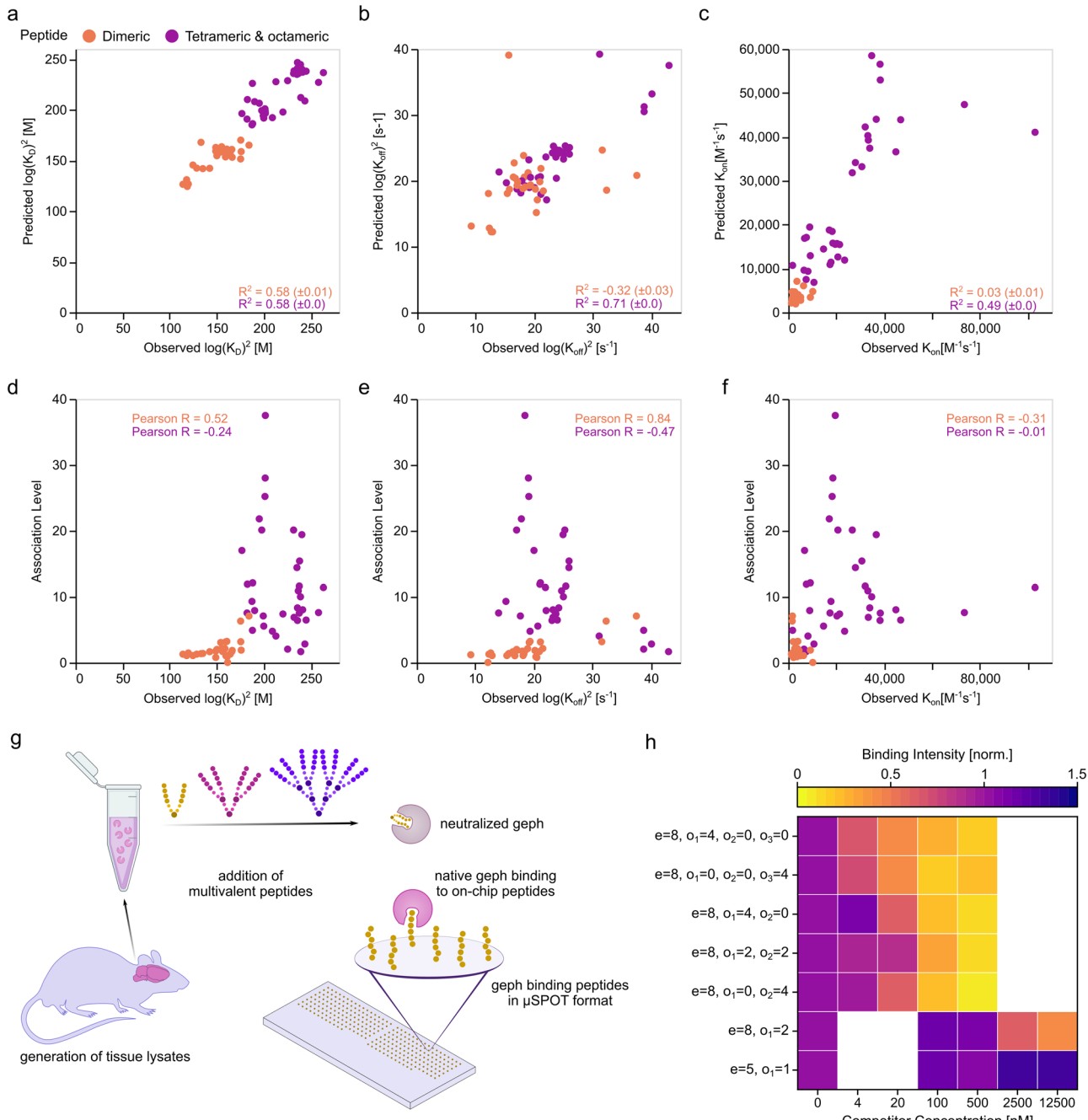

**Fig. 4 Multimer binding prediction and inhibition potency.** Measured $K_D$ values (**a**), off- (**b**), and on-rates (**c**) are plotted against predicted values in a leave-one-out cross-validation. Note the high correlation between predicted and obtained $K_D$ values. The obtained association levels are plotted against the observed $K_D$ values (**d**), off- (**e**), and on-rates (**f**). Note the low correlation between association levels and other kinetic parameters. **g** Schematic representation of µSPOT peptide microarrays, harbouring geph-binding peptides as cellulose conjugates. Native geph from mouse brain lysates was preincubated with multimeric peptides to neutralize geph-binding to on-chip peptides. **h** Normalized geph binding intensity to GlyR β-derived on-chip peptides in µSPOT format in the presence of varying competitor concentrations. Native geph binding to on-chip peptides was resolved by antibody detection and chemiluminescent readout. Note that tetrameric and octameric peptides effectively neutralized geph binding at lower concentrations than dimeric peptides. Data are presented as mean of $n = 2$ experiments. Source Data are provided as Supplementary Data 1.

FPS setup provided insights into the off-rates of these higher-architecture binders, which could be explained by the higher distance between the immobilized target protein in the heliX system compared to the distance on Ni-NTA biosensors in BLI, excluding complex re-binding effects on the biosensor surface. In addition, measurements of smaller and lower affinity dimeric peptides suffered from a poor signal-to-noise ratio in BLI (Supplementary Fig. 7), whereas FPS measurements provided superior signal-to-noise ratios largely independent of ligand size. In terms of resource consumption, FPS was on par with TRIC-based measurements and vastly outperformed both BLI and ITC. Yet, in our specific system, an inverse dependence of the observed on-rate on the employed analyte concentration was found (Supplementary Fig. 13), indicating that it's required to probe selected analytes at multiple concentrations before subjecting an array of varying compounds to a single-concentration screen and validate

selected hits in complementary biophysical methods such as ITC. A similar trend towards higher apparent on-rates when using increasing peptide concentrations was observed in BLI (Supplementary Fig. 14). This suggests that multimeric peptides, independently of the measurement method and protein immobilization, associate to the dimeric target protein in a concentration dependent manner. Another possible limitation in FPS are low signal-to-noise ratios when screening libraries with small compound size. This could be addressed by competition FPS setups with displaceable fluorescent compounds to further boost the signal amplitude[35].

Importantly, this study identified peptide-based binders with avidity enhanced inhibition capacity towards the ex vivo derived native protein. We anticipate that high-throughput FPS measurements in tandem with automated approaches for ligand synthesis will aid in similar projects advancing the rational optimization of effectors with unnatural building blocks[38] and other multimeric effectors, including multivalent protein-carbohydrate interactions[39]. The resulting high avidity binders will expand the toolbox of versatile chemical biology probes to advance our understanding of protein function and localization e.g. by manipulating receptor clustering[40] or by selective cell targeting with conjugated payloads[41].

## Methods

Unless otherwise stated, amino acids and reagents were purchased from either Iris Biotech or Carl Roth. All solvents were purchased from commercial sources and used without further purification.

**Automated solid-phase peptide synthesis**. µSPOT peptide arrays[42] were synthesized using a MultiPep RSi robot (CEM GmbH, Kamp-Lindford, Germany) on in-house produced, acid-labile, amino-functionalized, cellulose membrane discs containing 9-fluorenylmethyloxycarbonyl-β-alanine (Fmoc-β-Ala) linkers (average loading: 130 nmol/disc – 4 mm diameter)[43]. Synthesis was initiated by Fmoc deprotection using 20% piperidine (pip) in dimethylformamide (DMF) followed by washing with DMF and ethanol (EtOH). Peptide chain elongation was achieved using a coupling solution consisting of preactivated amino acids (aas, 0.5 M) with ethyl 2-cyano-2-(hydroxyimino)acetate (oxyma, 1 M) and N,N′-diisopropylcarbodiimide (DIC, 1 M) in DMF (1:1:1, aa:oxyma:DIC). Couplings were carried out for 3 × 30 min, followed by capping (4% acetic anhydride in DMF) and washes with DMF and EtOH. Synthesis was finalized by deprotection with 20% pip in DMF (2 × 4 µL/disc for 10 min each), followed by washing with DMF and EtOH. Dried discs were transferred to 96 deep-well blocks and treated, while shaking, with sidechain deprotection solution, consisting of 90% trifluoroacetic acid (TFA), 2% dichloromethane (DCM), 5% H$_2$O and 3% triisopropylsilane (TIPS) (150 µL/well) for 1.5 h at room temperature (rt). Afterwards, the deprotection solution was removed, and the discs were solubilized overnight (ON) at rt, while shaking, using a solvation mixture containing 88.5% TFA, 4% trifluoromethanesulfonic acid (TFMSA), 5% H$_2$O and 2.5% TIPS (250 µL/well). The resulting peptide-cellulose conjugates (PCCs) were precipitated with ice-cold ether (0.7 mL/well) and spun down at 2000 × g for 10 min at 4 °C, followed by two additional washes of the formed pellet with ice-cold ether. The resulting pellets were dissolved in DMSO (250 µL/well) to give final stocks. PCC solutions were mixed 2:1 with saline-sodium citrate (SSC) buffer (150 mM NaCl, 15 mM trisodium citrate, pH 7.0) and transferred to a 384-well plate. For transfer of the PCC solutions to white coated CelluSpot blank slides (76 × 26 mm, Intavis AG), a SlideSpotter (CEM GmbH) was used. After completion of the printing procedure, slides were left to dry ON.

**Preparative peptide synthesis**. Standard solid-phase peptide synthesis with Fmoc chemistry was applied, shortly, 2-chlorotrityl resin (1.6 mmol/g) was swollen in dry DCM with 2 eq. of N,N-Diisopropylethylamine (DIEA). Then, the desired aa (1 eq) and the orthogonally protected Boc-Gly-OH (1eq) were loaded. Boc-Gly-OH reduces resin loading in order to prevent aggregation of the elongating peptide chain. After ON reaction, the resin was capped with MeOH and washed with DCM and DMF. Deprotection and conjugation cycles followed, where 20% pip solution in DMF was used to deprotect, and after washes, the peptide chain was elongated by adding aa (4eq.) with oxyma (4eq.) and DIC (4eq.). Coupling efficiency was monitored by measuring the absorption of the dibenzofulvene–pip adduct after deprotection. The peptides were cleaved from the resin using a cocktail of 90.5% TFA, 4% H$_2$O, 3% TIPS 5% thioanisole, 2.5% 1,2-Dithiothreitol for 2 h at rt. The peptides were precipitated and washed twice with ice-cold ether, then purified with high-performance liquid chromatography (HPLC), and analyzed by liquid chromatography-mass spectrometry (LCMS) (Supplementary Table 3).

Unmodified peptides synthesized in 2 µmol scale were bought from Intavis Peptide Services (SKU: 90.215) with a free N-terminal amino end and C-terminal amide group and were used for FPS and BLI measurements without further purification. Crude peptide purity was assessed by LC-MS similar to preparatively synthesized peptides (Supplementary Table 4).

**Preparation of mouse tissue lysates**. Whole mouse brains were obtained from C57BL/6 J mice at >4 weeks of age and immediately flash-frozen in liquid N$_2$. Before lysis, whole mouse brains were weighed and cut into four pieces along the horizontal and vertical axis. To prepare one lysate, two diagonally opposite pieces were transferred into a 1.5 mL reaction tube (Sarsted). Lysis was carried out on ice in 500 µL HEPES lysis buffer (20 mM HEPES, 100 mM KCH$_3$COO, 40 mM KCl, 5 mM MgCl$_2$, 5 mM DTT, 1 mM PMS, 5 mM EDTA, 1% Triton X-100, 1% complete EDTA-free protease inhibitor cocktail (Roche) (all v/v)), by hand crushing the brain material with a metal pestle in a 1.5 mL reaction tube. Lysis was completed by 1 min sonification on ice with a Sartorius Labsonic M Sonificator at 20% amplitude with care to avoid heating the suspensions. Finally, Lysates were centrifuged for 15 min at 17,200 × g and 4 °C. The SN was subsequently collected, transferred to a new 1.5 mL reaction tube, flash-frozen in liquid N$_2$ and stored at −80 °C until use.

**Microarray binding assay**. µSPOT slides were blocked by incubation with 2.5 mL 5% (w/v) blotting grade milkpowder (MP, Carl Roth) in PBS for 60 min at ~70 revolutions per minute (rpm) and RT. Afterwards, slides were incubated with 0.8% (v/v) mouse brain lysate 5% MP in 1 × PBS for 15 min before slides were washed with 3 × 2.5 mL 1 × PBS for 1 min. To label native geph for detection, the slides were incubated with 2.5 mL of a 1:5,000 diluted primary antibody (anti-gephyrin (3B11), SynapticSystems) in 5% MP in 1 × PBS for 15 min, after which the slides were washed with 3 × 2.5 mL 1 × PBS for 1 min. Afterwards, the slides were incubated with a secondary HRP-coupled Anti-mouse antibody (31430, Invitrogen) in 5% MP in 1 × PBS for 15 min, after which the slides were washed with 3 × 2.5 mL 1 × PBS for 1 min. Peptide binding was detected through chemiluminescent detection (Lowest Sensitivity, 30 s exposure time) after application of 200 µL of SuperSignal West Femto Maximum Sensitive Substrate (Thermo Scientific) per slide using a c400 imaging system (Azure).

For on-chip peptide competition, native geph was preincubated with the indicated peptides in 5% MP in PBS for 30 min on ice before being put on an array slide.

Binding intensities were evaluated using FIJI including the Microarray Profile addon (OptiNav). After background subtraction of the mean greyscale value of the microarray surface surrounding the spots, raw greyscale intensities for each position were obtained for the left and right sides of the internal duplicate on each microarray slide. The standard deviation (STDEV) between both sides was obtained using formula (1).

$$STDEV = \sqrt{\frac{\sum(x - \bar{x})}{n}} \qquad (1)$$

with

$n$ The total number of data points
$\bar{x}$ The mean intensity value

Afterwards, the raw intensities of all spots of interest were summed and normalized to the summed intensity of the condition without competitor peptide.

**Protein expression and purification**. GephE (gephyrin P2 splice variant residues 318–736) was expressed in Escherichia coli and purified in a two-step purification as described earlier[29, 44]. Concisely, the protein was purified using via Intein-tag (Chitin beads, New England BioLabs), and after self-cleavage the protein could be obtained by size-exclusion chromatography (SEC) column (HiLoad 16/600 Superdex 200 pg, GE Healthcare) on an ÄKTA explorer system (GE Healthcare). His-tagged gephE was produced similarly with the exception of purification on IMAC coloumns before SEC purification.

**Temperature related intensity change (TRIC) assays**. For the TRIC assay, 16-point affinity measurements with each peptide against a target complex in duplicates were performed on the Dianthus NT.23PicoDuo. The experiment was performed in a single Dianthus 384-microwell plate using an assay buffer of 1 × PBS, 2 mM reduced L-Glutathione and 0.1% Pluronic® F-127, pH 7.4. Target protein and tracer peptide was diluted to 40 nM gephE and 20 nM NN1D-Cy5 in assay buffer and incubated on ice for one hour to create the target complex. All peptides were first pre-diluted to 2 mM in assay buffer and subsequently, a 16 point, 1:1 dilution series of each peptide was performed with an electronic multichannel pipette to a final volume of 10 µl directly in the Dianthus plate. Afterward, each dilution was mixed with 10 µl target complex, resulting in 16-point dilutions series of the peptides with a final concentration from 1 mM to 30.52 nM in the assay with 20 nM gephE and 10 nM NN1D-Cy5. The plate was centrifuged for 30 s at 1000 × g and incubated at 25 °C for 30 min. The final measurement of the plate was performed at 25 °C where the fluorescence signal of the samples was measured for 1 s with the IR-laser off and for 5 s on, resulting in TRIC traces where the detected fluorescence values are displayed as the relative fluorescence over time and under influence of the IR-laser induced heating and normalized to a value of one. For further analysis of the assay, the fluorescent signal is again normalized by dividing

the fluorescence values after IR laser activation with the fluorescence values prior to the activation giving the normalized fluorescence $F_{norm}$ in ‰. For a competitions assay of this kind, the affinity is evaluated by $K_i$ values which are obtained by applying a Hill-fit to a plot of $F_{norm}$ vs. ligand concentration to determine an $EC_{50}$ value (Formula 2 and 3).

The affinity of the tracer peptide to gephE was determined in the same assay buffer as the TRIC experiments were performed. gephE was diluted to 1000 nM and subsequently a 16-point dilution series of the protein was performed directly in a Dianthus plate in triplicate to a final volume of 10 µL. The gephE dilutions were mixed directly with 10 µL 2 nM NN1D-Cy5 to a final volume of 20 µL at 1 nM NN1D-Cy5 with protein concentration between 500 and 0.015 nM. The samples were subject to the same Dianthus parameters as above but analysed with a $K_D$ fit for later use in the determination of $K_i$ values.

$$K_i = \frac{K_D}{2 - \gamma} \cdot \left( \frac{EC_{50}}{\frac{[T]_t}{\gamma} - \frac{K_D}{2 - \gamma} - \frac{[C]_t}{2}} - \gamma \right) \qquad (2)$$

with

$$\gamma = \frac{[T]_t + [C]_t + K_D - \sqrt{([T]_t + [C]_t + K_D)^2 - 4[T]_t[C]_t}}{2[C]_t} \qquad (3)$$

and

$[T]_t$ Final concentration of the target protein (gephE)

$[C]_t$ Final concentration of fluorescent tracer peptide (NN1D-Cy5) that is in competition with unlabelled peptide ligand in the assay

$K_D$ The determined $K_D$ between the fluorescent tracer and the target protein from a direct binding affinity measurement

$EC_{50}$ The $EC_{50}$ obtained from the above-described competition assay between the unlabelled peptide ligand with the target complex

**Isothermal titration calorimetry (ITC).** ITC measurements were performed using an ITC200 (MicroCal) at 25 °C and 1000 rpm stirring. PBS pH 7.4 was used as the standard solvent. Specifically, 40 µL of a solution 200 µM of dimeric, or 100 µM of tetrameric compounds was titrated into the 200 µL sample cell containing 20 µM GephE. In each experiment, a volume of 2.5 µL of ligand was added, resulting in 15 injections and a final molar ratio between 1:0.5 (tetrameric compounds) and 1:1 (dimeric compounds). The dissociation constant ($K_D$) and stoichiometry (N) were obtained by data analysis using NITPIC, SEDPHAT, and GUSSI[45].

**Biolayer interferometry (BLI).** BLI measurements were carried out using the ForteBio Octet RED96 system. The chamber temperature was kept constant at 25 °C with a plate agitation speed of 1000 rpm. Briefly, Ni-NTA-coated biosensors were dipped into 200 µL of a 200 nM His-GephE solution (in a kinetic buffer (KB): 1 × PBS with 0.1% (w/v) BSA, 0.05% (v/v) Tween20, 2 mM GSH) for protein immobilization. The loaded sensors were moved to solutions containing various concentrations (200–0.781 nM) of dimeric, tetrameric and octameric peptides solubilized in KB to obtain the association curve. After the 180–300 s association step, the sensors were moved to KB to obtain the dissociation curve. A buffer only condition with a loaded biosensor was used as a reference for background subtraction. The association and dissociation curve were fitted with the ForteBio Biosystems Data Analysis high-throughput Software (local fitting algorithm, 1:1 model).

**Preparation of protein-DNA conjugates.** GephE was covalently coupled via its primary amines to the 5′end of ssDNA (cNL-A48, ligand strand) (coupling kit HK-NHS-1, Dynamic Biosensors, Martinsried, DE). In short, a heterobifunctional crosslinker is reacted with a DBCO modified DNA which yields an NHS activated DNA ester[46]. After removing excess crosslinker, the activated DNA is incubated with the protein, where the most reactive amine within the target protein will form an amide bond with the modified DNA strand. The protein-DNA conjugate was purified from the free protein and free DNA using the proFIRE® system (Dynamic Biosensors, Martinsried, DE)[46, 47]. The purification gives a good first impression of the status of the protein after conjugation and can be used as quality control of the protein sample to be immobilized onto the surface (for the chromatogram, see Supplementary Fig. 15). The embedded Data Viewer software provides protein-DNA conjugate purity and concentration based on the chromatogram. The yield of the gephE-DNA (1:1 ratio) is sufficient for approximately 300 chip functionalizations, considering a chip density of 100% and a ligand concentration of 100 nM. After liquid nitrogen freezing, the conjugates were stored at a concentration of 500 nM in PE40 buffer (10 mM $Na_2HPO_4$/$NaH_2PO_4$, 40 mM NaCl, 0.05 % Tween20, 50 µM EDTA, 50 µM EGTA) at −80 °C and were freshly thawed before each experiment.

**Chip functionalization.** All switchSENSE experiments were performed on a dual-color heliX$^+$ instrument using a standard heliX Adapter Biochip (ADP-48-2-0, Dynamic Biosensors, Martinsried, DE), in which single-stranded DNA (anchor strands) are covalently attached to the chip surface and the proteins are attached via flexible linkers to the DNA and are therefore able to rotate and bend quickly,

thereby exposing a large portion of their surfaces to the dye and reducing the effect of the immobilization on the measurement. Each chip is equipped with 2 gold electrodes (or spots), with different DNA anchor strands. Herein, we used spot 1 as measurement spot with the conjugated target protein (gephE-DNA) and spot 2 as real time referencing (only DNA), in order to monitor possible unspecific binding of the peptides on the anchor DNA and/or gold electrodes. Firstly, the conjugate gephE-DNA (ligand strand) was preincubated with the complementary ssDNA carrying the Gb fluorophore (adapter strand), for 20 min at RT upon shacking (600 rpm). Secondly, the whole ligand construct was immobilized on the biochip via hybridization of complementary anchor strand (for a schematic representation, see Supplementary Fig. 16). The chip was regenerated and freshly functionalized before each measurement series. For chip regeneration, the double stranded DNA nanolevers were denatured by disrupting the hydrogen bonds between base pairs using a high-pH regeneration solution (HK-REG-1, Dynamic Biosensors). The conjugate is washed away while the covalently attached single-stranded nanolevers remained on the surface and could be reused for a new functionalization step. Using FPS mode, a DNA-based biochip can be regenerated up to 50 times.

**Fluorescence proximity sensing (FPS) mode—switchSENSE interaction analysis.** Interaction analysis was performed in fluorescence proximity sensing (FPS) mode with a constant voltage of −0.4 V applied, which forces the surface-tethered DNA into a fixed angle. When the protein analyte binds to the DNA target, it affects the average distance of the fluorescent label from the fluorescence-quenching gold surface. Besides the change in DNA orientation, a change in close proximity to the fluorescent dye or direct interaction of the protein with the fluorescent dye lead to measurable changes in the fluorescence intensity. In the FPS measurements, the series of peptides were being flushed at specified concentrations over the two electrodes of the biochip. When the peptide reaches the target protein (gephE) present in spot 1, we observed an increase in the fluorescence signal on the timescale of seconds. Hence, the concentration jump itself may be considered instantaneous, and the time dependence of the fluorescence signal directly reflects the protein-peptide kinetics. After flushing out the peptide and replacing the bulk solution with pure buffer, only dissociation can take place. During measurements the sample tray containing the protein/peptide samples was set to 25 °C, as well as the experiment temperature on the biochip. Peptide samples were diluted and measured in PE140 buffer (10 mM $Na_2HPO_4$/$NaH_2PO_4$, 140 mM NaCl, 0.05% Tween20, 50 µM EDTA, 50 µM EGTA). Flow rate for association and dissociation reactions was set to 200 µL/min. The green LED power was set to 4. Experiment design, workflow and data analysis were performed with the heliOS software (Dynamic Biosensors, Matinsried, DE). The association and dissociation rates ($k_{on}$ and $k_{off}$), dissociation constants ($K_D$) and the respective error values were derived from a global single exponential fit model, upon double referencing correction (blank and real-time).

**SwitchSENSE relative size analysis—dynamic mode.** The hydrodynamic status/drag of gephE was investigated in a switchSENSE relative sizing experiment. The DNA levers' orientation and movement can be accurately controlled by applying alternating potentials to the gold surface where the nanolevers are immobilized. Due to the DNA's intrinsic negative charge, the nanolevers are attracted towards or repelled from the surface by applying a positive or negative potential, respectively. The motion is recorded in real-time via the fluorescence intensity of the nanolever layer. The fluorescence intensity depends on the distance between the fluorophore and the gold surface and hereby on the orientation of the nanolevers in relation to the gold surface. The distance-dependent fluorescence intensity is based on a distance-dependent, non-radiative energy transfer from the reporter fluorophores (on the nanolevers distal end) to the metal surface[48–51]. By alternating the applied surface potentials between an attractive and a repulsive potential at a frequency of 250 Hz, the DNA's intrinsic negative charge enables forced switching movement of the nanolevers. The nanolever movement happens on the microsecond scale, and can be fully resolved employing time-correlated single photon counting, with sampling events of 25 ns precision. To describe the DNA's motion between lying and standing states, the fluorescence intensity of the nanolever layer was recorded as a function of time, resulting in fluorescence response curves. The slope and form of the sigmoidal transition within the response curve is affected by the hydrodynamic drag of the nanolevers. As such, a nanolever carrying a protein cargo at its distal end experiences a notably larger hydrodynamic drag, corresponding to the hydrodynamic radius of the sampled protein. Switching measurements of double stranded DNA molecules with and without protein (gephE) were performed in a 10 mM Tris-HCl buffer (pH 7.4), containing 40 mM NaCl, 0.05% Tween20, 50 µM EDTA and 50 µM EGTA. The switching dynamics of DNA-protein complexes gradually slow down with increasing protein size. By comparing the measured upward switching fluorescence response of a sample under investigation (DNA-protein conjugate) to bare DNA and DNA protein conjugate standards, the hydrodynamic protein size can be set in relation to the protein standards and hereby estimated.

**Machine learning.** We employed Snakemake v6.9.1 using Python v3.8.5 to develop the machine learning workflow[52]. First, we removed all sequences with no available $K_D$ values, on-, or off-rates. Moreover, we used the median values for duplicated

sequences, i.e., repeated measurements. Afterward, we log-square-transformed $K_D$ and $k_{off}$ to retain issues with floating-point arithmetic. Specifically, we applied the FunctionTransformer from scikit-learn v1.0 using $log(x)^2$, with log being the natural logarithm[53]. We encoded the peptides using the amino acid composition (AAC)[31] and the linker sequence through the one-hot encoding. Thus, we introduced three binary representations to transform the linker into a machine-readable format and assigned the actual linker (J) to [1, 0] and the spacer (O) to [0, 1]. Since the model requires a fixed-length input, we also introduced gaps, denoted as [0, 0].

The AAC encoding counts the number of all amino acids concerning the total sequence length[54]:

$$f(t) = \frac{N(t)}{N} \qquad (4)$$

$N(t)$ denotes the number of amino acids $t$, $N$ refers to the peptide length, and $f(t)$, finally, is the composition of $t$[54]. The resulting matrix X contains 79 peptides represented by 20 proteinogenic amino acids and a binary vector of length 14, thus, 34 features. Note that we removed all AAC features with zero variance before model training.

Afterward, we used the Random Forest Regressor with default arguments. We verified the model employing leave-one-out cross-validation (LOOCV), i.e., we trained $k$ models using $k - 1$ peptides to predict the $k$-th peptide. For model evaluation, we computed the correlation coefficient $R^2$, which is defined as

$$R^2 = 1 - \frac{\sum_{i=1}^{n}(y_i - \widehat{y_i})^2}{\sum_{i=1}^{n}(y_i - \underline{y})^2} \qquad (5)$$

Specifically, $y_i$ is the $i$-th observed $K_D$ value, on-, or off-rate, $\widehat{y_i}$ is the $i$-th predicted $K_D$ value, on-, or off-rate, and $\underline{y}$ is the average $K_D$ value, on-, or off-rate. To score the correlation between the association level and $K_D$ values, on- and off-rates, we utilized Pearson's product-moment correlation coefficient:

$$Pearson\,R = \frac{\sum_{i=1}^{n}(x_i - \underline{x})(y_i - \underline{y})}{\sqrt{\sum_{i=1}^{n}(x_i - \underline{x})^2}\sqrt{\sum_{i=1}^{n}(y_i - \underline{y})^2}} \qquad (6)$$

We used the implementations provided by the scikit-learn library. Finally, we conducted a 1000-fold bootstrapping for the total $R^2$ and confidence interval (CI) calculation.

**Statistics and reproducibility**. Analysis was performed using OriginPro 2021 9.8.0.200 (OriginLab, Northampton, MA) and error bars represent mean ± SD unless otherwise noted. Replicates are technical, representing independent measurements. Number of repetitions varied depending on methodology and is defined in the figure legends.

**Reporting summary**. Further information on research design is available in the Nature Research Reporting Summary linked to this article.

## Data availability

The datasets generated during and/or analysed during the current study are available from the corresponding author on reasonable request.

## Code availability

The code generated during the current study is available under: https://github.com/spaenigs/fluorescence-proximity-sensing.

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

## Acknowledgements

We thank Sonja Kachler for her excellent technical assistance and Ricarda Berger for assistance with the hydrodynamic friction evaluation. This work was funded by the DFG (DFG MA6957/1-1) to H.M.M. and C.S. and by the Bundesministerium für Wirtschaft und Energie (BMWi) in the project MoDiPro-ISOB (16KN0742325) to A.S., N.A., I.B., S.S., R.S., D.H., W.S. This publication was supported by the Open Access Publication Fund of the University of Wuerzburg.

## Author contributions

Conceptualization: H.M.M., C.S.; Methodology: C.S., A.S., N.A., I.B., S.S.; Software: S.S., D.H.; Formal analysis: C.S., A.S., N.A., I.B., S.S.; Investigation: C.S., A.S., N.A., I.B.; Writing—original draft: C.S., H.M.M.; Writing—review & editing: A.S., S.S., N.A., I.B., R.S., D.H.; Visualization: C.S., S.S.; Supervision: H.M.M., D.H., R.S., W.S.; Project Administration: H.M.M., R.S., W.S., D.H.; Funding acquisition: H.M.M., R.S., W.S., D.H.

## Funding

## Competing interests

The authors declare the following competing interests: A.S. and R.S. are employed at Dynamic Biosensors which commercializes the heliX® system for FPS measurements. NA, IB and WS are employed at Nanotemper which commercializes the Dianthus® system for TRIC measurements. The other authors declare no competing interests.
