## [Peer Review File · Communications Biology]

Reviewers' comments:

Reviewer #1 (Remarks to the Author):

In the research article "Multivalent Binding Kinetics Resolved by Fluorescence Proximity Sensing" Schulte et al., report the development of a high throughput assay for measuring binding affinities between a target protein and a large library of peptides. The assay uses a "Fluorescence Proximity Sensing (FPS)" method which reports on fluorescence signal changes triggered by environmental changes around the dye – in this case, an altered environment upon binding of a peptide ligand to the target protein molecule. The authors then combine this procedure with automated peptide synthesis, interrogate multivalent peptide libraries and report that binding affinities scale with the valency of ligand for a particular target protein.

The idea behind the work is interesting and it is potentially useful since it adds a new technique for high-throughput identification of specific protein-interaction partners from large libraries of potential candidates. However, I found the work lacking in crucial details and control experiments which limit my enthusiasm to some degree.

I list my major concerns below:

1. The authors report the work as a general method for studying protein-protein interactions. However, only one target protein has been used to establish the FPS method. The dye environment in presence of different proteins can vary widely and it may turn out that other target proteins do not produce similar large variations in analogous FPS assays. It is therefore not clear how generally applicable this method would be. Demonstration of the basic FPS method with one or two target proteins of varying size would make the work stronger - or else they should at least discuss this caveat.
2. The authors claim that one of the strong suits of the FPS method is that both the target protein and the ligand library are unlabeled and as a result, we do not need to worry about the "functional integrity" of the interacting molecules. However, this is not the case because the target protein is immobilized via a reaction between primary amines in the protein and an NHS ligand on a nucleic acid strand. It could result in a pretty large change in the functional integrity of a protein.
3. It is stated that GephE was linked to one of the strands via NHS ester linkage. The chemistry used here is not clear. Do the authors mean the linkage is done via amines on the protein? If, yes do the authors use a specific amine (e.g., the N-terminal -NH₂), or do they use a non-specific method? If a non-specific method has been used how would the number of amines on a protein affect the efficiency of the FPS assay described here? This could also affect the reported target protein consumption values.
4. It is also possible that attachment via different amines results in different orientations of the target protein with respect to the dye. Is it likely the dynamic range of the assay would be affected for other protein targets due to this factor?
5. It is also possible there could be reactive amines in the binding site of the target protein. How would it affect the assay?
These points need to be discussed when discussing the generalizability of this method.
6. I could not find some basic control experiments to check the evolution of the baseline fluorescence signal:
 - a. A full experiment without GephE monitoring fluorescence signal in presence of the peptides.
 - b. A full experiment with GephE but no peptide.
 - c. An experiment with a mutant GephE which alters the ligand binding site.
7. The annotations in Figure 3 are very difficult to follow. The authors need to find a better way to make this understandable for the general reader.
8. What happens when the octameric peptides are used in the experiment in Fig. 4?
9. There are numerous typos and errors throughout the manuscript. Please find and fix it.

Reviewer #2 (Remarks to the Author):

This study investigates the effect of peptide length and different multimeric presentation on the kinetics and affinity of binding to gephyrin. The authors compare the utility of fluorescence proximity sensing with three other biophysical techniques. Screening over a hundred multimeric peptides they discover that higher on-rates coupled with a flexible core are the key determinants for high affinity binding as the degree of peptide valency increases. The correlation between higher valency and high affinity binding was validated from competition experiments with ex vivo derived material. This data has allowed machine learning to test prediction of the kinetic and binding parameters.

Overall, this is a well thought out and executed study whose conclusions will be of worth to a wide readership due to the interest in multi-valent binding and the development of therapeutics especially those targeting protein-protein interfaces. I have several major concerns that needs to be addressed before acceptance:

1 line 271 "... Yet, in our specific system, an inverse dependence of the observed on-rate on the employed analyte concentration was found (Supplementary Figure 10), indicating that it's required to probe selected analytes at multiple concentrations before subjecting an array of varying compounds to a single-concentration screen and validate selected hits in complementary biophysical methods such as ITC."

As the increase in affinities with valency are dependent on an increase in the on-rate constants, it could just be there is a difference in concentration dependency between different valencies. While the authors show two examples in the supplementary data, they need to show that the concentration dependency is not orders of magnitude different between different classes. It would be good for them to comment why they think there is such a dependency. A simple explanation may be there is self-association between the peptides reducing the free concentration that is able to associate. Have the authors checked for self-association or aggregation? Why does it not seem to be an issue with BLI or TRIC?

2. line 108 "Compared to ITC measurements, which can be considered the gold standard as they quantify directly and label-free in solution (Figure 2A)"

This system is outside the parameters of optimal ITC measurement. Measuring low nM affinity with protein in the μM range leads to a step-function in the titration as seen in Supplementary Fig 8 with only one or two points in the transition region leading to inaccurate K_d determination. I think the authors should state these limitations. The authors should also address the observed low stoichiometry in these binding experiments. Does this indicate that the measurement of peptide concentration was inaccurate, or the free concentration is lower, if as suggested above, as a possible explanation for the concentration dependence on the on-rate constants? Or with higher valency peptides do multiple proteins bind a single peptide?

Minor points:

1. I found Figure 1 A confusing. I don't think this represents any of the peptides tested but what is possible. Either a simplified diagram or better one representing the different classes of valencies with the different numbering schemes labelled may aid the general reader. The detailed chemistry could be moved to Supplementary. Or maybe more explanation in the text.

2. line 112 "ligand concentrations that did for effective dissociation of tetramers"
Change to ligand concentrations that prevented effective dissociation of tetramers

3. line 295 insert "at a " before concentration

4. very minor point: make sure all "Figure" in the supplementary data are capitalised.

Reviewer #3 (Remarks to the Author):

In the article titled "Multivalent Binding Kinetics Resolved by Fluorescence Proximity Sensing" by Clemens Schulte et al., the authors have attempted to synthesize a series of peptides, some of them of multi-dentate nature, and attempted to correlate their binding affinities, on-rate and off-rate with valency using fluorescence proximity assay. It is mostly a method development paper and has little relevance to any biological questions.

Although the subject matter may be of interest to some investigators, there are several questions regarding the methodology. Probably the biggest question in my mind is what is the valency in the bound state? The peptides may have the potential to bind in multi-valent mode, but do they actually bind in such a mode. In fact, K_d comparisons do not provide any support to that idea (For example, see Figure 3B) The K_d s between dimers, tetramers and octamers barely vary 2-3 fold. I would expect the K_d s to be orders of magnitude different. The same pattern is seen in ITC data (Figure 2A). It is important to determine K_d s of tetrameric peptides whose two epitopes are scrambled sequences and compare them with that of the pure dimer.

If I understood the technique correctly, the K_d obtained by this method is an amalgamation of microscopic dissociation constants (binding of a single epitope). The proximity assay will only give the step that affects the probe proximity. If the binding is a multi-step process, only the first association step may be reported. A corollary is that if the third and fourth epitope binding is significantly slower than the first two binding, it may be totally transparent to FPS as the probe is already in the final state due to the presence of the peptide in the proximity of the target. I feel the authors should address these questions before the article can be accepted.

Other issues

What is the subunit composition of bacteria purified gephyrin splice variant used in the study? This question is important because the multivalent binding will depend on the number of subunits present. This needs to be determined. The flipside of this question is what is the subunit composition of the immobilized gephyrin? That is the species on which this study was carried out.

Minor comments

Page 3, line 40: What do the authors mean by "affine" in relationship with inhibitors? It is usually used in Mathematics.

We thank the reviewers for their detailed revision and the constructive feedback. We have adjusted our manuscript according to the suggestions and performed additional experiments as requested by the reviewers to improve the quality of our manuscript.

Answers to the reviewers in blue

Reviewer #1 (Remarks to the Author):

In the research article “Multivalent Binding Kinetics Resolved by Fluorescence Proximity Sensing” Schulte et al., report the development of a high throughput assay for measuring binding affinities between a target protein and a large library of peptides. The assay uses a “Fluorescence Proximity Sensing (FPS)” method which reports on fluorescence signal changes triggered by environmental changes around the dye – in this case, an altered environment upon binding of a peptide ligand to the target protein molecule. The authors then combine this procedure with automated peptide synthesis, interrogate multivalent peptide libraries and report that binding affinities scale with the valency of ligand for a particular target protein.

The idea behind the work is interesting and it is potentially useful since it adds a new technique for high-throughput identification of specific protein-interaction partners from large libraries of potential candidates. However, I found the work lacking in crucial details and control experiments which limit my enthusiasm to some degree.

We thank the reviewer for the overall positive evaluation and the guidance on how to clarify our points in the manuscript. In the revised version of the manuscript, we added missing details and control experiments to substantiate the value of FPS in quantifying protein-protein interaction affinities und evaluating kinetics of multivalent binders.

I list my major concerns below:

1. The authors report the work as a general method for studying protein-protein interactions. However, only one target protein has been used to establish the FPS method. The dye environment in presence of different proteins can vary widely and it may turn out that other target proteins do not produce similar large variations in analogous FPS assays. It is therefore not clear how generally applicable this method would be. Demonstration of the basic FPS method with one or two target proteins of varying size would make the work stronger - or else they should at least discuss this caveat.

We thank the reviewer for this comment on the general applicability. In the revised version of the manuscript, we now added content that substantiates the general applicability of the FPS principle for quantification for monofunctional ligands including protein-interactions in a non-high-throughput fashion and further elaborate on setting up FPS assays.

Successful examples include measurements with human serum albumin (Wenskowsky et al., ChemBioChem, 2020, 15), an IgG fusion antibody binding to HER2 (Kast et al., Nat. Commun., 2021, 12), the transcription factor FOXP2 binding to DNA (Häußermann et al., Angew. Chem. Int. Ed., 2019, 58), and thrombin binding aptamer (Ponzo et al., Molecules, 2019, 24), which differ significantly in size and composition.

The revised manuscript now also summarizes design principles that are shared among successful FPS applications. E.g., attaching the dyes via flexible linkers to facilitate quenching or anti-quenching (as for gepHE) effects upon ligand binding.

2. The authors claim that one of the strong suits of the FPS method is that both the target protein and the ligand library are unlabeled and as a result, we do not need to worry about the “functional integrity” of the interacting molecules. However, this is not the case because the target protein is immobilized via a reaction between primary amines in the protein and an NHS ligand on a nucleic acid strand. It could result in a pretty large change in the functional integrity of a protein.

We thank the Reviewer for focusing our attention on the possible problems resulting from improper protein immobilization. In the revised version of the manuscript, we highlight the importance to check the structural integrity of the immobilized protein. In this context we described the necessary workflows and added the respective experimental data that substantiate the integrity of protein after immobilization and the complete absence of unspecific binding. Specifically, we conducted a series of additional measurements in the same setup to determine the hydrodynamic status by measuring the sample induced LAG value of the protein (2.1) and to demonstrate the absence of unspecific binding using non-binding protein variants (2.2):

2.1 The determined sample induced LAG value of gepH, in comparison to those of a series comprised of monomeric, globular proteins that were measured under the same conditions, corresponds to a higher molecular weight of a dimer. These data have now been added to the supplementary material of the manuscript. The corresponding methods section has also been updated with a brief description of the methodology.

Supplementary figure 1: switchSENSE relative size analysis – Dynamic Mode and Lag Value of GephE. (A) Theoretical explanation of the high frequency dynamic electrical switching mode which probes the hydrodynamic radius (as function of the friction) of analyte molecules and serves to determine the size and shape of biomolecules. The hydrodynamic radius of the ligand/sample (or hydrodynamic diameter) adds additional drag to the nanolevers when these are pushed and pulled through solution. Thus, the larger the ligand/sample, the slower the motion/switching speed. Therefore, the hydrodynamic radius of the conjugated protein (gephE) can be estimated by comparing the time-resolved fluorescence motion curve of gephE with switching curves of other proteins of known weight and size. (B) Theoretical explanation of the dynamic lag value. At a given time, the DNA nanolever with the ligand (sample) moves a shorter distance than the control DNA nanolever (control). It lags in distance and fluorescence behind the control, due to additional friction created by the ligand. The absolute dynamic lag corresponds to the area between the imaginary zero-drag nanolever (dotted grey line) and either the control (blue line) or the sample curve (pink line). In accordance with increasing hydrodynamic friction, small ligands correspond to fast switching and large ligands to slower switching ($r \sim$ dynamic lag). The sample-induced lag is the area between the control and the sample curve, i.e. the difference between the curve integrals. (C) Fluorescence motion curves of all measured proteins (including two independently functionalized batches of gephE – (1) and (2)) and the respective control with an empty ligand strand. (D) Zoomed-in version of panel (C). (E) Absolute LAG values of each protein and the respective control yield the sample-induced LAG value. (F) Sample-induced LAG values of each measured protein are plotted against the respective molecular weight. Note that the sample-induced LAG values measured for gephE (red) do not correlate with the rest of the LAG values that were measured for monomeric, globular proteins. This suggests that gephE is dimeric in the immobilized form.

2.2 Absence of unspecific binding, aggregation or other non-stoichiometric effects beyond the expected protein-protein interaction were excluded by additionally conducted FPS experiments with non-binding gephE variants (see point 6 c)).

We further changed our original statement that highlighted that “FPS neither requires direct fluorescent labelling of the ligand nor the analytes, thereby avoiding disturbance of their functional integrity or other dye-mediated artefacts”. The new description now better highlights that the protein still requires immobilization. Although misleading, we trust the reviewer will agree that the FPS method can be grouped with other so-called “label-free methods” including surface-based biosensor methods (e.g. SPR, BLI) which also require immobilizing proteins on the surface.

Old sentence:

Importantly, FPS neither requires direct fluorescent labelling of the ligand nor the analytes, thereby avoiding disturbance of their functional integrity or other dye-mediated artefacts.

New sentence:

Importantly, FPS neither requires direct fluorescent labelling of the ligand nor the analytes, thereby avoiding dye-mediated artefacts.

3. It is stated that GephE was linked to one of the strands via NHS ester linkage. The chemistry used here is not clear. Do the authors mean the linkage is done via amines on the protein? If, yes do the authors use a specific amine (e.g., the N-terminal -NH₂), or do they use a non-specific method? If a non-specific method has been used how would the number of amines on a protein affect the efficiency of the FPS assay described here? This could also affect the reported target protein consumption values.

We thank the reviewer for pointing out this omission. We now added a description of the immobilization procedure:

In short, a heterobifunctional crosslinker is reacted with a DBCO modified DNA which yields an NHS activated DNA ester (Reinking et al., STAR Protocols, 2021, 2). After removing excess crosslinker, the activated DNA is incubated with the protein, where the most reactive amine within the target protein will form an amide bond with the modified DNA strand. After incubation, a purification step is carried out using anion exchange chromatography (proFIRE[®]). Here, unreacted DNA and protein are removed. In this and our earlier studies, we did not observe significant differences between individually labeled protein batches. The updated version of the manuscript now furthermore states that the target protein consumption may vary depending on the efficiency of the immobilization strategy used. Yet, we did not observe significant differences in protein consumption (chip functionalization in FPS) between different batches of modified gepHE.

4. It is also possible that attachment via different amines results in different orientations of the target protein with respect to the dye. Is it likely the dynamic range of the assay would be affected for other protein targets due to this factor?

Prompted by the reviewer we have revised the description of our immobilization protocol: We now highlight that the target protein is attached via a flexible linker to the DNA and is therefore able to rotate and bend quickly, thereby exposing a large portion of the surfaces to the dye and reducing the effect of the immobilization on the measurement.

In addition, the measurements of the hydrodynamic status of the immobilized gepHE, that have been added to the revised supplementary information of the manuscript (see point 2 above), show that individually immobilized protein batches display identical hydrodynamic statuses and

the observation that measurements are consistent with the label-free measurements indicate a comparably minor effect of the applied labelling and immobilization strategy.

5. It is also possible there could be reactive amines in the binding site of the target protein. How would it affect the assay? These points need to be discussed when discussing the generalizability of this method.

We now elaborate further on this point. In our work, the complementary biophysical methods confirm that immobilization for FPS did not affect protein activity. We also shortly discuss the alternatives to amine coupling. Namely, thiol coupling, which exploits free thiol-reactive site, like cysteines, or the site-specific covalent conjugation of his-tagged proteins to DNA, which uses the amine in proximity of the his-tag. active site protection strategies as demonstrated for kinases using ATPyS and MgCl₂, or standard capture strategies (like his-tag, avi-tag, twin-strep-tag, Fc A-G-tag and GFP-tag) are also possible.

6. I could not find some basic control experiments to check the evolution of the baseline fluorescence signal: a. A full experiment without GephE monitoring fluorescence signal in presence of the peptides. b. A full experiment with GephE but no peptide.

We thank the reviewer for highlighting this very important point and apologize for the omission. The revised version of our manuscript now also describes the real-time reference that was always included to control for fluorescent signal in absence of the target protein upon peptide injection (see below the revised version of figure 1 B). Specifically, the measurements were performed in two spots at the same time, one with the protein and an additional one without the protein, but the same DNA sequence and same dye. Thereby effectively allowing to control for unspecific binding of the analyte to the biochip template, understanding how the dye interacts with the protein, and how the fluorescence signal evolves over time. The second spot is always recorded, and it has been used as reference for the peptide screening.

Figure 1 B: schematic representation of FPS measurements. The receptor-binding domain of the neuronal scaffolding protein gephyrin (gephE) is immobilized on the ligand strand via an NHS coupling. The binding of unmodified, multimeric peptides during the association phase is detected by a change in fluorescence of green dye b. Note that a real-time reference on spot 2 is used to control for unspecific binding or influence on the fluorophore

In the revised version of the manuscript, we further added an example of the parallel raw signals to the supplementary information (see below), which shows high specificity of the tetrameric peptide ($e=8$, $o_1=2$ $o_2=2$) towards gephE (panel A), while no binding has been detected on spot 2, where only the DNA is located (panel B).

Supplementary figure 4: Control conditions in FPS measurements. To account for a potential influence of the peptide on the fluorophore, a condition without immobilized ligand (gephE, panel (A)) is measured in parallel on spot 2 of the measurement chip (B) for each peptide concentration. In addition, a condition without peptide is used to monitor changes in baseline fluorescence.

In addition, for the 1:1 kinetic experiment we measured and used as reference the “blank run” (or concentration 0), in which gephE is present on the ligand strand, but only buffer without peptide is injected. An example is shown in the previous figure (figure above - “blank”).

c. An experiment with a mutant GephE which alters the ligand binding site.

We addressed this with additional experiments evaluating two different non-binding gephE variants (gephE F330A and P713E) against 3 different tetrameric peptides ($e=8, o_1=4, o_2=0$; $e=8, o_1=2, o_2=2$; and $e=8, o_1=0, o_2=4$). Both gephE point variants are known to significantly decrease the binding affinity of gephE towards the GlyR β subunit and derived peptides that were used in this study (both gephE variants were first described in Kim et al., EMBO J., 2006, 25). As becomes apparent from the plots below, while the gephE wildtype is consistently showing a clear binding upon peptide injection, the two variants do not show any binding upon injection. Multiple concentrations have been tested up to 100 μ M, however for clarity of the graph we just present one, which has been also used to measure the kinetics of gephE (see figure SI 6). These extra analyses were now added to the supplementary information.

Supplementary Figure 2: Control experiment with two non-binding *gephE* point variants. Binding of three tetrameric peptides with varying architecture (A) $e=8, o_1=4, o_2=0$, (B) $e=8, o_1=2, o_2=2$, (C) $e=8, o_1=0, o_2=4$ to *gephE* wildtype and two non-binding point variants (P713E, F330A, first described in (Kim et al., 2006)) was evaluated in FPS. Note that neither one of the point variants exhibit binding to the tetrameric peptides in contrast to wildtype *gephE*.

7. The annotations in Figure 3 are very difficult to follow. The authors need to find a better way to make this understandable for the general reader.

We thank the reviewer for bringing this to our attention. To clarify the presentation of the data in figure 3, it was modified in the following way:

- 1) Annotations for the three different subgroups of multimeric peptides (dimers, tetramers and octamers) were added above the respective group.
- 2) Special annotations for on- and off-rate, which are derived from the association (assoc.) and dissociation (dissoc.), respectively, within the upper left graph (panel A)) make it easier to interpret the lower panels D and E, where the on-rate is plotted against the off-rate.
- 3) We added additional explanations to the figure legend

New version of figure 3:

8. What happens when the octameric peptides are used in the experiment in Fig. 4?

To answer this question, we conducted and added to the manuscript (revised figure 4 panel G and H) a competition assay with two octameric peptides ($e=8, o_1=4, o_2=0, e_3=0$ and $e=8, o_1=0, o_2=0, e_3=4$), which represent the octamer with the highest and lowest affinity, respectively. The octameric peptides neutralize on-chip native geph binding at least on par with the tested tetrameric peptides.

New Figure 4 panel G and H:

9. There are numerous typos and errors throughout the manuscript. Please find and fix it.

We carefully revised the text and corrected the typos and text errors in the updated version of the manuscript.

Reviewer #2 (Remarks to the Author):

This study investigates the effect of peptide length and different multimeric presentation on the kinetics and affinity of binding to gephyrin. The authors compare the utility of fluorescence proximity sensing with three other biophysical techniques. Screening over a hundred multimeric peptides they discover that higher on-rates coupled with a flexible core are the key determinants for high affinity binding as the degree of peptide valency increases. The correlation between higher valency and high affinity binding was validated from competition experiments with ex vivo derived material. This data has allowed machine learning to test prediction of the kinetic and binding parameters.

Overall, this is a well thought out and executed study whose conclusions will be of worth to a wide readership due to the interest in multi-valent binding and the development of therapeutics especially those targeting protein-protein interfaces. I have several major concerns that needs to be addressed before acceptance:

We thank the Reviewer for the encouragement and the focused suggestions on how to clarify our points.

1 line 271 "... Yet, in our specific system, an inverse dependence of the observed on-rate on the employed analyte concentration was found (Supplementary Figure 10), indicating that it's

required to probe selected analytes at multiple concentrations before subjecting an array of varying compounds to a single-concentration screen and validate selected hits in complementary biophysical methods such as ITC."

As the increase in affinities with valency are dependent on an increase in the on-rate constants, it could just be there is a difference in concentration dependency between different valencies. While the authors show two examples in the supplementary data, they need to show that the concentration dependency is not orders of magnitude different between different classes. It would be good for them to comment why they think there is such a dependency. A simple explanation may be there is self-association between the peptides reducing the free concentration that is able to associate. Have the authors checked for self-association or aggregation? Why does it not seem to be an issue with BLI or TRIC?

We thank the reviewer for focusing our attention on this point. Prompted by this comment we revisited our datasets and indeed identified a similar trend of concentration-dependent on-rates in the BLI data (now added as Suppl. Fig. 14). Here, it becomes apparent that the same two peptides that exhibited a concentration-dependent on-rate in FPS also do so in BLI (see below), however yielding a lower K_D value in higher concentrations (inversely to the trend observed in FPS, supplementary figure 13). Since TRIC measurements do not resolve on- or off-rates of the binding events, such a dependency cannot be observed.

Regarding potential self-association of the multimeric peptides at higher concentrations, we did not observe any unusual heat signatures in ITC (see raw heat signatures of dimeric and tetrameric peptides below) that would indicate self-association at increasing peptide concentrations (peptides were titrated into gepH solution).

In the ITC experiments, high μM concentrations of dimeric and tetrameric peptides were employed, similarly to the FPS titration experiment in supplementary figure 13. We performed this FPS titration experiment to check for potential biases of FPS towards higher peptide concentrations. Accordingly, the screening and affinity determinations shown in the main figures were conducted at low and constant peptide concentrations which are consolidated by the in-solution methods TRIC and ITC. The revised version of the manuscript highlights the need to carefully control for systematic concentration-dependent biases in the surface-based techniques (FPS and BLI).

2. line 108 "Compared to ITC measurements, which can be considered the gold standard as they quantify directly and label-free in solution (Figure 2A)"

This system is outside the parameters of optimal ITC measurement. Measuring low nM affinity with protein in the μM range leads to a step-function in the titration as seen in Supplementary Fig 8 with only one or two points in the transition region leading to inaccurate K_D determination. I think the authors should state these limitations.

We now describe our ITC measurements as "high c" measurements (conducted at upper edge of the dynamic concentration range of ITC). Thus, accurately resolving enthalpy and stoichiometry but contributing only few data points to the determination of the exact dissociation constant.

The authors should also address the observed low stoichiometry in these binding experiments. Does this indicate that the measurement of peptide concentration was inaccurate, or the free concentration is lower, if as suggested above, as a possible explanation for the concentration dependence on the on-rate constants? Or with higher valency peptides do multiple proteins bind a single peptide?

We thank the reviewer for highlighting the observed stoichiometry and apologize for the omission. In the revised version of the manuscript, we now interpret the observed stoichiometry in the context of the expected binding mode. The measured stoichiometry in ITC is consistent with expected binding of one dimeric peptide binding two gepH proteins at once ($n=0.5$). Further, the observed stoichiometry of $n=0.25$ is in line with one tetramer simultaneously binding four gepH proteins.

Minor points:

1. I found Figure 1 A confusing. I don't think this represents any of the peptides tested but what is possible. Either a simplified diagram or better one representing the different classes of

valencies with the different numbering schemes labelled may aid the general reader. The detailed chemistry could be moved to Supplementary. Or maybe more explanation in the text.

To simplify the depiction of the different classes of peptides tested in this study and to make the variations within the classes more apparent, we now show the dimeric, tetrameric and octameric in a separated way (see the revised version of figure 1 A) below).

2. line 112 "ligand concentrations that did for effective dissociation of tetramers" Change to ligand concentrations that prevented effective dissociation of tetramers

The text was changed accordingly.

3. line 295 insert "at a " before concentration

The text was changed accordingly.

4. very minor point: make sure all "Figure" in the supplementary data are capitalised.

The figure labels in the supplementary are now capitalized similar to the labels in the main text.

Reviewer #3 (Remarks to the Author):

In the article titled "Multivalent Binding Kinetics Resolved by Fluorescence Proximity Sensing" by Clemens Schulte et al., the authors have attempted to synthesize a series of peptides, some of them of multi-dentate nature, and attempted to correlate their binding affinities, on-rate and off-rate with valency using fluorescence proximity assay. It is mostly a method development paper and has little relevance to any biological questions.

Although the subject matter may be of interest to some investigators, there are several questions regarding the methodology. Probably the biggest question in my mind is what is the valency in the bound state? The peptides may have the potential to bind in multi-valent mode, but do they actually bind in such a mode. In fact, K_d comparisons do not provide any support to that idea (For example, see Figure 3B) The K_d s between dimers, tetramers and octamers barely vary 2-3 fold. I would expect the K_d s to be orders of magnitude different. The same pattern is seen in ITC data (Figure 2A). It is important to determine K_d s of tetrameric peptides whose two epitopes are scrambled sequences and compare them with that of the pure dimer.

We thank the reviewer for the interest in the methodology. From the stoichiometry of the ITC measurements (supplementary figure 11) it can be concluded that the dimeric peptides bind two gepH proteins ($n=0.5$) and the tetrameric peptides bind four gepH proteins ($n=0.25$).

The observation that the K_d values (dissociation rates) do not vary as strongly as the association rates between dimers, tetramers and octamers can be explained by the fact that the target protein gepH is dimeric in its immobilized form during the measurements. In the revised version of the manuscript, we now consolidated this point by performing measurements of the hydrodynamic status of the immobilized gepH (see third point). Consequently, any excess valency over the two binding sites of the gepH dimer contributes almost exclusively to an improved on-rate (up to 100-fold) whereas off-rate effects are moderate (up to three-fold). Thus, lending an explanation for the moderate global affinity gains for the higher valency binders.

A control experiment with tetramers containing two scrambled binding epitopes would require a conceptionally different synthesis strategy which we consider outside of the scope of this study. Yet, the obtained kinetic SARS does resolve clear trends for the on- and off-rates in relation to the overall multivalent binder architecture indicating that the orientation of the ligands via the multivalent architecture matters. We thus assume that tetrameric peptides with two scrambled epitopes would bind similarly to dimeric peptides.

If I understood the technique correctly, the K_d obtained by this method is an amalgamation of microscopic dissociation constants (binding of a single epitope). The proximity assay will only give the step that affects the probe proximity. If the binding is a multi-step process, only the first association step may be reported. A corollary is that if the third and fourth epitope binding is significantly slower than the first two binding, it may be totally transparent to FPS as the probe is already in the final state due to the presence of the peptide in the proximity of the target. I feel the authors should address these questions before the article can be accepted.

We thank the reviewer for pointing out this aspect about the FPS phenomenon when applied to multivalent interactions. The revised version of the manuscript now addresses these questions together with the underlying citations. In brief, FPS can be expected to resolve complex binding events, especially if the on-rates of the two binding events are differ. An example from the literature where more than one phase in the association was resolved was added to the main text: (Kast et al., Nat. Comm., 2021, 12). As we observed only a single

phase in the association and dissociation of the respective peptides, we conclude that our multivalent interaction does not progress via a multi-step binding event. Additionally, gephE is only present as a dimeric species in its immobilized form (see third point) in FPS measurements. Since this is also the case in ITC and BLI measurements, where no indications of a multi-step interaction were observed (no delayed or broadened heat signatures in ITC or biphasic curves in BLI), the idea that the interaction between gephE and the multimeric peptides is taking place in a single association and dissociation phase is further supported.

Other issues What is the subunit composition of bacteria purified gephyrin splice variant used in the study? This question is important because the multivalent binding will depend on the number of subunits present. This needs to be determined. The flipside of this question is what is the subunit composition of the immobilized gephyrin? That is the species on which this study was carried out.

We thank the reviewer for bringing up the important question of composition and multivalency of the immobilized protein in the FPS. The revised manuscript now highlights a universal approach to directly assess information on the multimeric state of the immobilized protein, by carrying out a modified measurement within the same system that was also used for the FPS measurements. Using high frequency dynamic electrical switching we probed the hydrodynamic radius as function of the friction of our immobilized protein to estimate its size and shape. Here we exploit that proteins exert increasing drag to the nanolevers when these are pushed and pulled through solution with increasing protein hydrodynamic radius. Thus, the higher the weight of the immobilized protein, the slower the motion/switching speed. This measurement indicates that the immobilized gephE is indeed dimeric, in line with literature knowledge from X-ray crystallography and SAXS measurement in solution (Sander et al., Acta Crystallogr D Biol Crystallogr., 2013, 69). This conclusion can be drawn because in comparison to a series comprised of monomeric, globular proteins that was measured under the same conditions, gephE exhibits a LAG value that corresponds to the molecular weight of a dimer. The figure and corresponding methods section have been added to the manuscript. In summary, the revised manuscript now also provides a procedure to control and assess the relative size of the immobilized protein in situ before starting the kinetic binding quantification.

Supplementary Figure 1: switchSENSE relative size analysis – Dynamic Mode and Lag Value of GephE. (A) Theoretical explanation of the high frequency dynamic electrical switching mode which probes the hydrodynamic radius (as function of the friction) of analyte molecules and serves to determine the size and shape of biomolecules. The hydrodynamic radius of the ligand/sample (or hydrodynamic diameter) adds additional drag to the nanolevers when these are pushed and pulled through solution. Thus, the larger the ligand/sample, the slower the motion/switching speed. Therefore, the hydrodynamic radius of the conjugated protein (gephE) can be estimated by comparing the time-resolved fluorescence motion curve of gephE with switching curves of other proteins of known weight and size. (B) Theoretical explanation of the dynamic lag value. At a given time, the DNA nanolever with the ligand (sample) moves a shorter distance than the control DNA nanolever (control). It lags in distance and fluorescence behind the control, due to additional friction created by the ligand. The absolute dynamic lag corresponds to the area between the imaginary zero-drag nanolever (dotted grey line) and either the control (blue line) or the sample curve (pink line). In accordance with increasing hydrodynamic friction, small ligands correspond to fast switching and large ligands to slower switching ($r \sim$ dynamic lag). The sample-induced lag is the area between the control and the sample curve, i.e. the difference between the curve integrals. (C) Fluorescence motion curves of all measured proteins (including two independently functionalized batches of gephE – (1) and (2)) and the respective control with an empty ligand strand. (D) Zoomed-in version of panel (C). (E) Absolute LAG values of each protein and the respective control yield the sample-induced LAG value. (F) Sample-induced LAG values of each measured protein are plotted against the respective molecular weight. Note that the sample-induced LAG values measured for gephE (red) do not correlate with the rest of the LAG values that were measured for monomeric, globular proteins. This suggests that gephE is dimeric in the immobilized form.

Minor comments Page 3, line 40: What do the authors mean by “affine” in relationship with inhibitors? It is usually used in Mathematics.

The wording “affine” was referring to the affinity that describes the binding strength of the peptide towards its target protein. Since, as pointed out by the reviewer, this could be confusing to certain audiences, we changed the wording, and the sentence now reads:

“The development of selective PPI modulators with high target protein binding affinities is facilitated by biophysical technologies that enable the determination of binding parameters of large binder libraries with minimal sample requirements.

REVIEWERS' COMMENTS:

Reviewer #1 (Remarks to the Author):

The authors have provided reasonable answers to my queries. I would recommend the publication of this manuscript in Communications Biology.

Reviewer #2 (Remarks to the Author):

The authors have significantly revised the manuscript with additional data and changes to the text which has strengthened the work and increased the clarity of the methodology. All my concerns have been addressed in this revised version which is now, in my opinion, acceptable for publication

Reviewer #3 (Remarks to the Author):

The rebuttal is acceptable to me.